# Reliable RNA-seq analysis from FFPE specimens as a means to accelerate cancer-related health disparities research

Mitchell J. Frederick[1]*, Dannelys Perez-Bello[1], Pedram Yadollahi[1], Patricia Castro[2], Alan Frederick[3], Andrew Frederick[3], Rashid A. Osman[4], Fonma Essien[1], Imelda Yebra[1], Ashley Hamlin[1], Thomas J. Ow[5,6], Heath D. Skinner[7], Vlad C. Sandulache[1,8,9]*

1 Bobby R. Alford Department of Otolaryngology Head and Neck Surgery, Baylor College of Medicine, Houston, Texas, United States of America, 2 Department of Pathology & Immunology, Baylor College of Medicine, Houston, TX, United States, 3 Informagen, Wilmington, D.E., United States, 4 - Department of Biological Sciences, Vanderbilt University College of Arts and Science, Nashville, Tennessee, United States of America, 5 Department of Otorhinolaryngology-Head and Neck Surgery, Montefiore Medical Center, Bronx, New York, United States of America, 6 Department of Pathology, Montefiore Medical Center, Bronx, New York, United States of America, 7 Department of Radiation Oncology, UPMC Hilman Cancer Center, Pittsburgh, Pennsylvania, United States of America, 8 ENT Section, Operative CareLine, Michael E. DeBakey VAMC, Houston, Texas, United States of America, 9 Center for Translational Research on Inflammatory Diseases, Michael E. DeBakey Veterans Affairs Medical Center, Houston, Texas, United States of America

* vlad.sandulache@bcm.edu (VCS); Mitchell.Frederick@bcm.edu (MJF)

## Abstract

Whole transcriptome sequencing (WTS/ RNA-Seq) is a ubiquitous tool for investigating cancer biology. RNA isolated from frozen sources limits possible studies for analysis of associations with phenotypes or clinical variables requiring long-term follow-up. Although good correlations are reported in RNA-Seq data from paired frozen and formalin fixed paraffin embedded (FFPE) samples, uncertainties regarding RNA quality, methods of extraction, and data reliability are hurdles to utilization of archival samples. We compared three different platforms for performing RNA-seq using archival FFPE oropharyngeal squamous carcinoma (OPSCC) specimens stored up to 20 years, as part of an investigation of transcriptional profiles related to health disparities. We developed guidelines to purify DNA and RNA from FFPE tissue and perform downstream RNA-seq and DNA SNP arrays. RNA was extracted from 150 specimens, with an average yield of $401.8\,ng/cm^2$ of tissue. Most samples yielded sufficient RNA reads >13,000 protein coding genes which could be used to differentiate HPV-associated from HPV-independent OPSCCs. Co-isolated DNA was used to identify reliably define patient ancestry which correlated well with patient-reported race. Utilizing the methods described in this study provides a robust, reliable, and standardized means of DNA & RNA extraction from FFPE as well as a means by which to assure the quality of the data generated. Optimized RNA extraction techniques, combined with robust bioinformatic approaches designed to optimize data homogenization, analysis and biological validation can revolutionize our ability to transcriptomically profile large solid tumor sets derived from ancestrally varied patient populations.

**Data availability statement:** All data analyzed in the manuscript are included in the supplementary tables.

**Funding:** This work was supported by the National Institute of Dental and Craniofacial Research (R21DE032344- VCS), the National Institutes of Health (R01DE0323337, P50 CA097190, R01 DE028061- HDS), the Dan L Duncan Comprehensive Cancer Center (P30-CA125123), the Human Tissue Acquisition & Pathology Core Baylor College of Medicine, a Price Family Foundation Pilot Project to Develop Experimental Cancer Therapeutics for Underrepresented Populations (TJO), awarded by the Montefiore Einstein Comprehensive Cancer Center and an Administrative Supplement to Support Diversity Research awarded under NIH-NCI. 3U54CA274321-02S1 (TJO). The funders had no role in study design, data collection and analysis, decision to publish, or preparation of the manuscript.

**Competing interests:** The authors have declared that no competing interests exist.

## Introduction

Owing to technical advances in next generation sequencing (NGS) and widespread optimization of protocols and commercial kits for RNA extraction, whole transcriptome RNA expression profiling (i.e., RNA-seq) has become a ubiquitous research tool. Often, RNA-seq is performed using RNA isolated from fresh or snap-frozen samples [1–5]. While such investigations have provided insight into the biology of diseases, the availability of fresh or frozen samples can constrain study designs, create selection bias, and limit research endeavors that require long-term follow-up. This is unfortunate because many research institutions and laboratories have amassed large biobanks of archival specimens collected over decades during routine diagnosis, medical practice, or drug studies, which far outnumber collections of fresh/frozen tissue [6–8]. Therefore, the ability to reliably perform bulk RNA-seq profiling from archival formalin fixed paraffin embedded (FFPE) specimens would greatly expand scientific research [9]. Despite evidence that RNA-seq can successfully be performed using FFPE specimens of varying age [10–12], uncertainties regarding optimal RNA extraction procedures and the variability/ reliability of whole transcriptome expression data generated remain the biggest obstacles preventing more widespread acceptance and application [13,14].

One area that would benefit immensely from routine RNA-seq analysis of archival FFPE samples is health disparities research. In addition to socio-economic and cultural differences, there is evidence that disparities in clinical disease and outcome among minority populations could also have a biological basis [15–17]. If so, it is likely that known differences in single nucleotide polymorphisms (SNPs) underpinning ancestral differences at the genomic level could exert their phenotypic impact by influencing RNA expression levels of specific genes [15,18]. Moreover, demographic data collected through medical records can be imperfect [19] or altogether missing, making genomic ancestry a potentially more accurate method [20]. Due to this need, we describe optimized procedures to extract both RNA and DNA from FFPE tumor sections, guidelines for performing quality control (QC) of extracted nucleic acid, recommendations for specific RNA-seq and DNA SNP platforms, and suggest approaches for standardized data processing and data QC.

## Materials and Methods

### Ethics approval and consent to participate

All specimen collection, retrieval and analysis performed followed approval from Baylor College of Medicine and Michael E. DeBakey Veterans Affairs Medical Center (MEDVAMC) Institutional Review Boards. Specimens were accessed for research purposes on 01/08/2023.

Basic steps in our RNA data processing pipeline include: 1) Filtering out non-protein coding genes; 2) Excluding zero count data to calculate a $75^{th}$ percentile (UQ) read value for each sample; 3) Calculating the cohort median count upper quartile (MUQ) value; 4) Replacing any gene sizes $\leq 252$ bp with a minimum value of 252 (i.e., size of smallest known human gene) to prevent data distortion or a zero division error; 4) Calculating the global cohort gene size median (GMGS); 5) Normalizing each count value through dividing by the sample specific $75^{th}$ quartile and the individual gene size (either constant or sample specific); 6) Multiplying the quotient by the product of the MUQ x GMGS to bring numbers back to the scale of counts; 7) Performing a log2 transformation after adding back 0.01 to every value (Log2 [X + 0.01]); 8) Identifying and removing "technical" outlier samples using the average median absolute deviation (MAD) approach coupled with a downstream statistical method to detect MAD outliers; 9) Re-scaling the data so the distribution has a global median set to 7.0; and 10) Replacing negative log2 values with a zero, to avoid the pitfalls of artificially inflating standard deviations and fold changes associated with very low expression values. Importantly, because

the data are rescaled to median of 7 before replacing low expression values, zero replacements are handled consistently across data with different sequencing depth. Furthermore, it becomes feasible to select a single low expression threshold cut-off (e.g., Log2 ≤ 2) to suffice for every experiment across time and different sequencing depths. The protocol described in this peer-reviewed article is published on protocols.io at dx.doi.org/10.17504/protocols.io.6qpvr9kpzvmk/v1 (protocol.io) and is included for printing as supporting information file 1 with this article. Specific python codes and the supporting Gene_GC.csv input files are freely available from the Github repository at https://github.com/Mjfreder/RNA_normalize_qc.

## Results and discussion

### Isolation and quality control of RNA from FFPE specimens

Various publications have compared in-house and commercial kit-based procedures for optimally extracting RNA, DNA, or both from FFPE tissue [14,21–25]. Here, rather than re-investigate all the variables, our goal was to provide researchers with a detailed set of guidelines and instructions—all in one place— to successfully purify DNA and RNA from FFPE tissue and for downstream analyses like RNA-seq and DNA SNP arrays. Our procedures were extensively vetted and yielded robust results when applied by multiple individuals in our group, regardless of experience level. Although we make no claim that our methods produce superior results, we present them as best practices that could enhance the likelihood of experimental success using FFPE prepared tissue.

We opted to first isolate RNA using the AllPrep DNA/RNA FFPE kit from Qiagen and further process residual DNA pellets using the COBAS kit from Roche, with some modifications to the manufacturers' instructions. All RNA/DNA isolations began from pre-cut slides of oropharyngeal squamous cell carcinomas (OPSCC) prepared from archival blocks that ranged in storage time from 1 to 20 years. Alternatively, sections could be sliced directly into Eppendorf tubes and stored at room temperature in a desiccator prior to processing. FFPE slide sections were cut to 6 μm thickness; **Supplementary Table 1** in S1 Table summarizes the physical properties from 43 FFPE samples processed for RNA isolation and commercially sequenced, along with an additional 150 samples also analyzed for RNA yield but not sequenced. A breakdown of yields for the 43 sequenced specimens and for all 193 samples processed is provided in **Supplementary Table 2** in S1 Table. For the 43 sequenced samples, the average total RNA yield was 3629 ng (range of 485 ng to 13,247 ng), average area of tissue per slide was 2 cm$^2$, average number of slides used was ~9, average total area of tissue processed was 18 cm$^2$, and average yield per total area of tissue was 402 ng/cm$^2$ (median = 274 ng/cm$^2$). Most standard commercial RNA sequencing platforms can work with 200–500 ng RNA, albeit the true yields measured by fluorescent methods were usually half that determined by absorbance as discussed further below. Summary sample metrics for the entire cohort of 193 specimens were not fundamentally different from the subset of 43 selected for sequencing (**Supplementary Table 2** in S1 Table). We did not find a statistically significant relationship between how long tissue blocks had been stored and the RNA yields per square centimeter of tissue (Fig 1A).

We qualitatively and quantitatively compared the reliability of three different library/platforms for performing whole transcriptomic RNA-Seq from archival FFPE specimens (**Supplementary Table 1** in S1 Table). RNA extracted from a subset of FFPE specimens was processed through one commercial sequencing company that used either the TruSeq RNA exome kit from Illumina (recommended specifically for FFPE samples) for library preparation or the Illumina TruSeq Stranded kit followed by removal of noncoding RNA with the Ribo-zero kit. Another set of RNA samples was processed through a different vendor for massive analysis of cDNA ends (MACE-seq) that captures 3' ends of polyadenylated RNA, but samples were

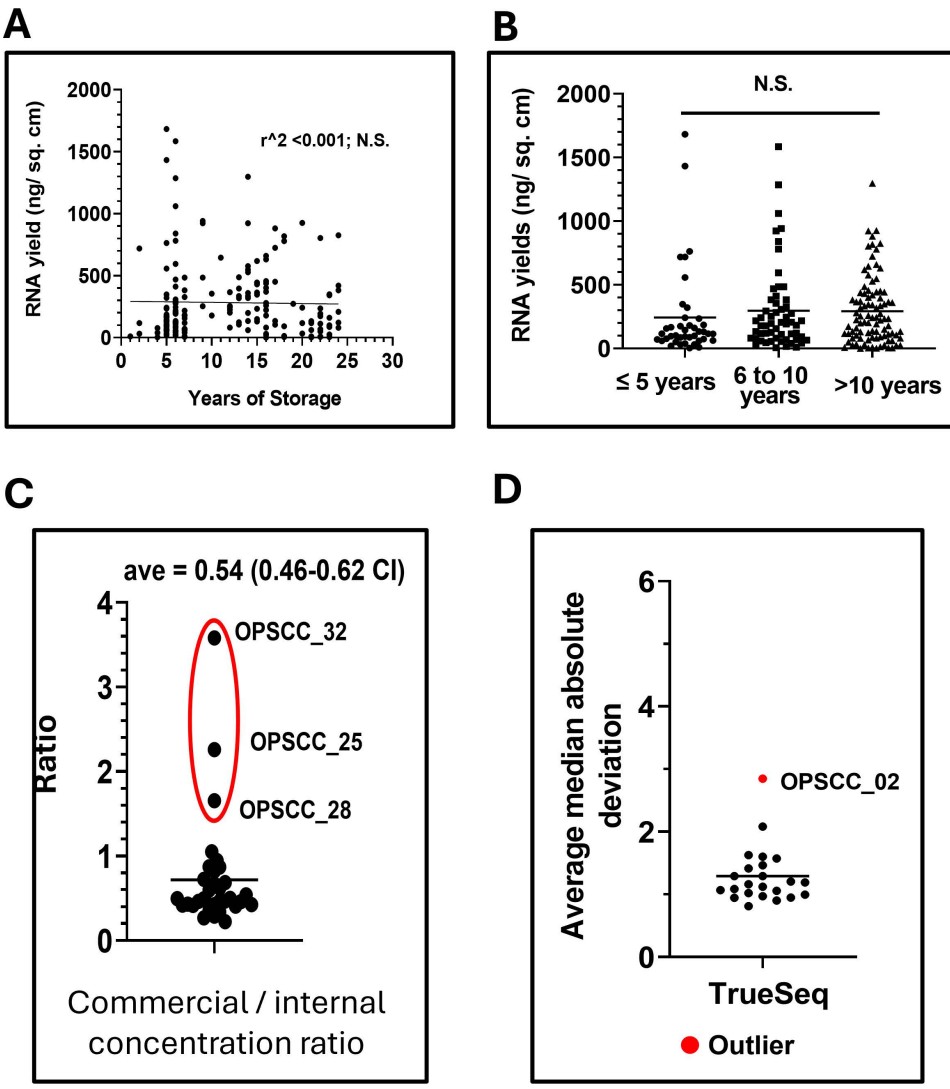

**Fig 1. RNA yields.** There was no correlation between RNA yields and relative age of the individual clinical specimens (A,B). C) Laboratory concentrations were approximately 50% of concentrations measured at the commercial vendor. D) Technical outlier detection using the MAD method, showing the average MAD values from the TrueSeq cohort, which cluster well except for outlier sample OPSCC_02 (Adj **P** < 0.0001, Supplementary Table 5 in S1 Table).

unavoidably stored at 4°C for several days during transit. However, MACE-seq potentially performs better than other platforms with partially degraded RNA [26,27].

We compared the RNA concentration measured in-house by absorbance to that obtained through fluorescence methods performed commercially (**Supplementary Table 3** in S1 Table, Fig 1B, S1 Fig), excluding samples that were improperly stored during transit. On average, in-house concentrations measured by absorbance were about half that obtained through fluorescence techniques (Fig 1B). Not surprisingly for FFPE specimens, RIN values were under the recommended value of 7 or higher and ranged from 1.2 to 2.5 (**Supplementary Table 3** in S1 Table). A more common metric for fragmented RNA is the DV200 (i.e., percentage of fragments ≥200 bp) which varied widely from 1.48% to 71.47% (**Supplementary Table 3** in S1 Table), with a median value of just 18.65, excluding the samples used for MACE-seq that were stored at warmer temperatures for an extended time.

## Recommendations for processing RNA-Seq data

There are many publications examining approaches for processing RNA-Seq data [28–32], each with its merits. Here, we present a user-friendly pipeline (S2 Fig) and Python code that normalizes RNA expression data in a standardized fashion from raw reads, puts gene expression values on the same scale regardless of the sequencing platform or depth of coverage, and controls for inter-sample variability. Furthermore, we incorporate procedures for consistent identification of technical outlier samples and minimizing the influence of low expression or zero values. These steps are not intended to substitute for batch corrections but do allow some gross comparisons and inferences to be made across experiments spanning different platforms. Nonetheless, our normalization methods and their outputs should be compatible with batch adjustment methods like the frequently used Combat algorithm [33,34]. Popular RNA-seq normalization methods include calculating fragments per kilobase of transcript per million reads (FPKM), or transcripts per million (TPM). We use an upper quartile (UQ) normalization approach known as FPKM-UQ, frequently adopted by The Cancer Genome Atlas (TCGA) project [35–37]. Each of these methods have been used successfully and they all control for gene size—which while not strictly necessary for analyzing differences in gene expression—should be employed whenever single sample gene set enrichment (ssGSEA) is a downstream analysis [9,38]. The UQ normalization in our pipeline divides gene expression values by a sample specific UQ value (i.e., 75th percentile excluding zeros), rather than a sample total, and has been shown by others to control well for inter-sample variability [39,40].

According to our data processing pipeline shown in S2 Fig the starting input file should contain meta data (i.e., combined sample data) with individual gene sizes and either raw counts (e.g., HTSeq-Count) or expected counts if RNA-Seq by Expectation Maximization (RSEM) was used. Gene sizes are constant across samples and derived purely through reference annotation when HTSeq-Counts are employed, but for RSEM the effective gene sizes will vary slightly from sample to sample. It's worth noting that many sequencing cores and companies already provide FPKM and TPM normalizations with the data [41,42]. However, such values should be used cautiously, because these normalizations frequently do not filter out non-protein coding genes in the process. It is not uncommon for the gene expression data provided by sequencing centers to include tens of thousands of non-protein coding genes with 40,000–60,000 "genes" listed when in fact there the true number of protein coding human genes is in the range of 19,000–20,000 (depending on whether immune specific genes are included). For easy reference, we provide our comprehensive list of 19,709 human protein coding genes (**Supplementary Table 4** in S1 Table) derived from the HGNC website along with their official symbols, crossmatched to their Entrez gene ID's, Ensembl IDs, and HGNC ID's.

A consequence of our normalization approach is that some downstream software packages for differential analysis of gene expression (DEG) that require raw data input and perform their own internal normalization, such as DESeq2 [43] or EdgeR [44], are not appropriate. DEG packages like Limma [45] or Voom [46] which accept pre-normalized data for linear modeling can be used. For simpler analyses between groups, we typically combine multiple T-testing with the Benjamini Hochberg procedure for controlling false discovery rate (FDR). The main advantage with our data processing pipeline is better control over what is happening during normalization.

## Analysis and comparison of RNA-Seq data sets generated from FFPE specimens using different sequencing approaches

The methods described here were developed to aid in comparing RNA-Seq data from multiple platforms, but they are appropriate for routine analysis of RNA-Seq experiments. The total

mapped reads or sequencing depth varied greatly between the different library preparations/ platforms we examined, with an average of roughly 17.3 million, 6.6 million, and 0.67 million for the TruSeq, TruSeq stranded, and MACE-seq approaches, respectively (**Supplementary Table 3** in S1 Table, S3 Fig). We examined the relationship between total mapped reads and reads mapped to protein coding genes (**Supplementary Table 3** in S1 Table, S3 Fig) and found best results with MACE-Seq (average = 88%), followed by TruSeq (average = 82%), compared to Stranded/Ribo-zero (average = 55%). Differences in sequencing depth between methods were reflected in the UQ values used during normalization (**Supplementary Table 3** in S1 Table).

The distributions of gene expression (i.e., median UQ normalized log 2 values) for each platform/cohort, along with the TCGA OPSCC RNA-Seq dataset similarly normalized, are shown in S4 Fig A,C,E, and G before global rescaling. To directly compare the behavior of RNA values it is helpful to re-scale each cohort to have a common global median. However, samples that represent technical outliers should first be removed to avoid data distortion. Outliers were identified using a two-step procedure based on the Median Absolute Deviation (MAD) approach. First, for every gene, MAD values were computed by taking the absolute value of a sample's gene expression minus the cohort median and delta values were then averaged across all genes to calculate a sample specific average MAD value. In the second step, a linear model was fit across all the sample average MAD values to generate residuals, which were normalized to calculate the probability (i.e., p-value) of observing each sample's average MAD score using the cumulative distribution function with an FDR = 0.01 for multiple testing. Samples OPSCC_02 (TrueSeq RNA exome), OPSCC_32 and 33 (Stranded RNA/Ribo-zero), OPSCC_34 and 35 (MACE-Seq) were all identified as outliers (**Supplementary Table 5** in S1 Table, FDR < 0.0001, Fig 1C, S5 Fig) and tended to have some of the lowest 75th PCTL raw counts relative to other samples in their respective cohorts.

The global medians of each data set (excluding outliers) were then used to re-scale cohorts to obtain distributions with a new and common global median of 7. Essentially, each normalized log2 gene expression value was modified by adding (7- initial global median) to obtain rescaled or adjusted values, except that gene expression values corresponding to true zeros were not adjusted. Histograms for the newly rescaled cohorts were generated (S4 Fig B,D,F,H). After this global adjustment, low expression values could be treated uniformly across cohorts. To minimize data distortion and noise around low expression genes, all negative log2 expression values were set to zero. For downstream applications like DEG that require testing multiple genes and p-value corrections it is helpful to minimize the number of statistical tests performed or genes analyzed. One advantage of our normalization procedure that rescales all cohorts to the same global median of 7 is that a universal low expression threshold can be set. We recommend filtering out genes where the maximum average log2 expression of any group ≤2, which typically results in anywhere from 12,000–16,000 genes being tested (e.g., **Supplementary Table 5** in S1 Table). Finally, all re-scaled log2 values <0 were uniformly substituted with zeros, to minimize analysis artifacts associated with small values.

## RNA sample quality metrics driving successful sequencing

As mentioned, RNA RIN values are of little use for assessing RNA isolated from FFPE samples. After examining the different sequencing metrics described above, we re-considered which, if any, RNA sample QC measurements might be useful to predict good quality sequencing results. TapeStation readouts for representative samples sequenced with the Stranded library plus Ribozero approach (Fig 2) or the TruSeq library (S6 Fig) illustrated several differences among samples.

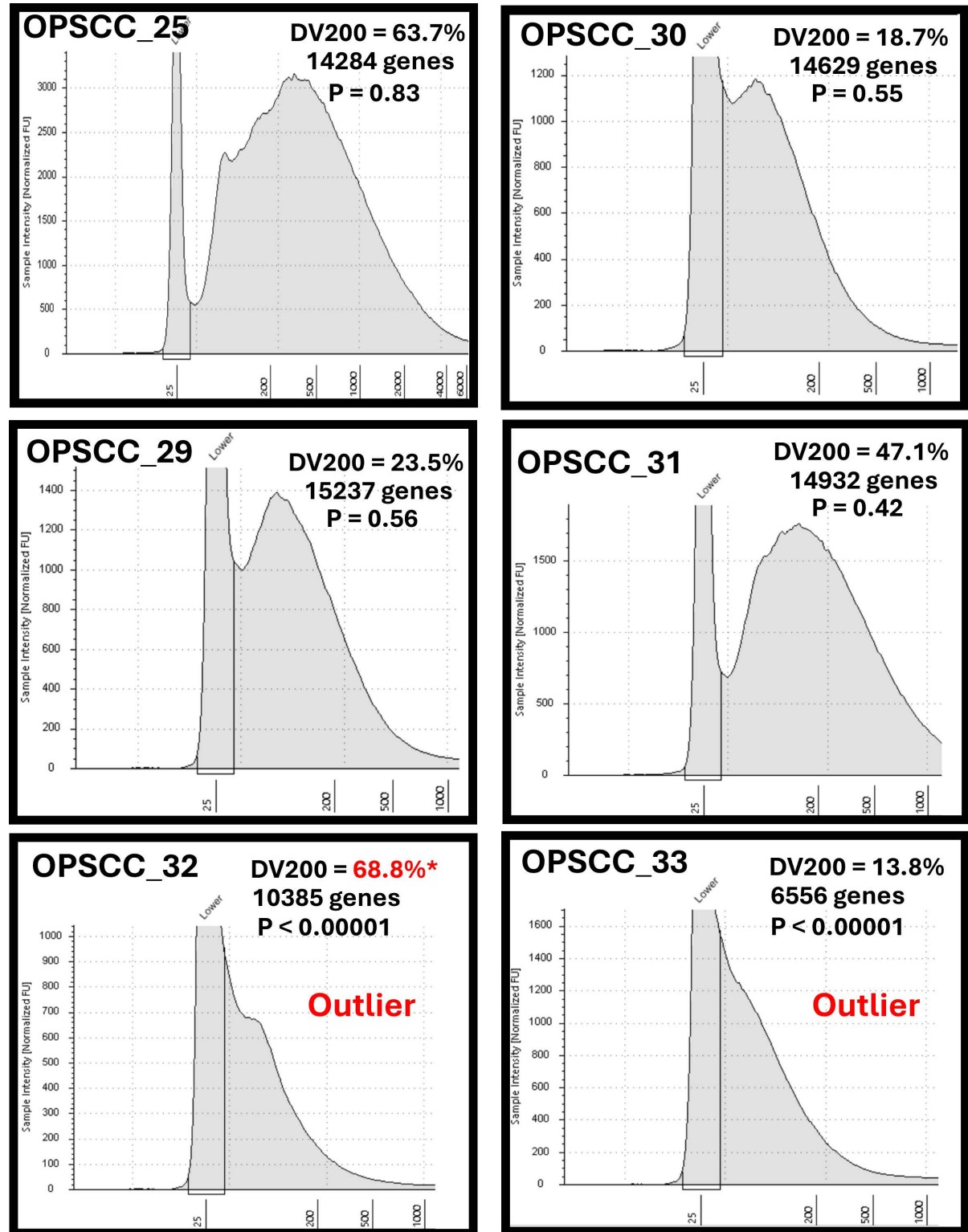

**Fig 2. Relationship between DV200 and biological richness of gene expression.** DV200 values and RNA fragment size are shown for representative samples along with the relative number of usable genes and outlier associated p-values from Supplementary Table 5. * denotes potentially erroneous DV200 reading.

Samples with large DV200 (e.g., percent of RNA molecules ≥200 bp) values, such as OPSCC_25 (DV200 = 63.7%) and OPSCC_31 (DV200 = 47.1%) yielded over 14,000 genes having log2 expression values ≥2 and were not among the outliers. Even samples with considerably lower DV200 values like OPSCC_09 (DV200 = 13.7%) or OPSCC_13 (DV200 = 19.9%) also yielded a substantial number of genes expressed above the low threshold cut-off. Outliers like OPSCC_02, OPSCC_32, and OPSCC_33, on the other hand, visibly had some of the lowest DV200 values (<14%). However, several samples like OPSCC_01 and OPSCC_16 also had DV200 values below 10%, but performed reasonably well with over 13,000 genes each expressed above the low threshold. These latter samples had very subtle shoulders visible in their TapeStation curves that distinguished them from the outlier OPSCC_02 sample in the TruSeq cohort. Generally, the samples used in the Stranded/Ribo-zero library were of better quality based on TapeStation curves, with more a greater proportion of RNA fragments in the 100 bp range. Collectively, our experience suggests that although a high DV200 percentage is very likely to produce good reads, samples with peaks or shoulders in the 100 bp range may also generate acceptable sequencing data.

## Biological Reliability of RNA-seq data from FFPE specimens

We found that wholesale examination of correlations among all protein coding genes between the TCGA cohort and our datasets was not particularly informative and seemingly good correlations existed even when tested against highly unrelated cancer types like pancreatic tumors. Very likely, the majority of genes do not contribute to biological variation and serve as a buffer when included in such analyses. Furthermore, if subsets of genes expected to be more highly expressed in a tissue specific manner were chosen for cross-dataset correlations using cohort medians, results were disappointing. By randomly modeling the cohort makeups, we concluded that this likely stems from the high variability introduced to cohort medians when analyzing small sample sets.

We theorized a more robust approach for examining biological fidelity of RNA-Seq datasets would be to look for known or expected patterns of gene co-expression, considering both positive and negative correlations. Clinically, OPSCCs can be divided by the presence or absence of high-risk human papillomavirus (HPV) as a cancer driver, which is reflected in very different biology and treatment outcomes [47–52]. The incidence of HPV-associated OPSCC has been steadily increasing in the last few decades and as of 2021, ~75% of cancers occurring in this anatomical subsite are associated with HPV [6,7]. We previously identified that 51 out of 78 TCGA OPSCC were HPV-associated based on re-analyzing publicly available whole exome sequencing data, and used this to define genes differentially expressed between HPV-associated and HPV-independent OPSCC after normalizing gene expression for the TCGA cohort (**Supplementary Table 6** in S1 Table). Next, we selected genes that showed ≥3-fold significant (FDR < 0.1) difference up or downregulation based on HPV status in the OPSCC TCGA cohort (S7 Fig, S6 Table) and filtered out immune genes to focus on differences in tumor biology, leaving 855 differentially regulated genes. Theoretically, these genes should show a similar pattern of expression in our cohorts, based on the assumption that both HPV-associated and HPV-independent tumors are expected. In reality, some amount of data noise is expected in our datasets due to platform differences or smaller cohort sizes. Therefore, we examined the co-correlation of these 855 differentially expressed genes in our largest FFPE cohort, the TruSeq library data set that included 20 samples.

Co-correlation coefficients from the 855 differentially expressed genes calculated using the TruSeq cohort are listed in **Supplementary Table 7** in S1 Table and hierarchical clustering was used to identify modules of genes with patterns of co-correlation or anti-correlation (S8 Fig), with directionality of fold change expected from the TCGA cohort annotated. It is worth

noting that genes with increased or decreased expression in the TCGA cohort according to HPV status were clearly segregating in the TruSeq FFPE cohort (S8 Fig), which is already indicative that biology was being strongly preserved. We selected genes in clusters 1 and 3 (collectively 598 out of 855 genes) to see if they could predict HPV status of our FFPE specimens. Because we do not have the HPV status of the samples used in our study and the p16 (CDKN2A) immunohistochemistry (IHC) sometimes used as surrogate was not available for all samples, we relied on CDKN2A mRNA levels. Overexpression of CDK2NA RNA, which also drives p16 protein detection by IHC, results from a negative feedback loop in HPV-associated cancers with diminished Rb and highly active E2F transcription factor. Using the OPSCC TCGA cohort we demonstrated that HPV-associated tumors abundantly overexpress CDKN2A RNA with an average of 34-fold elevation (**Supplementary Table 6** in S1 Tables) that is highly significant ($p < 0.0001$, S7 Fig B).

We used the 598 DEGs to cluster samples from all 3 of our FFPE cohorts (**Supplementary Tables 8**–10-**10** in S1 Tables) to predict their HPV status (Fig 3–4, S9 Fig) based on the patterns of their gene expression. In the TruSeq cohort (Fig 3), two main gene clusters were found which were enriched or depleted for genes previously determined to be differentially expressed in the TCGA cohort based on HPV status. The segregation of these genes was highly significant ($p < 0.00001$) indicating strong preservation of tumor biology. This was additionally validated by significant elevation of CDKN2A RNA among the FFPE specimens predicted to be HPV-associated ($p < 0.005$, Fig 3B), despite the fact that CDKN2A had been purposely removed from the list of genes for clustering. Highly similar results were found for the Stranded/Ribo-zero and Mace-Seq datasets, regarding segregation of the HPV-signature genes, although genes associated with HPV-independent tumors formed two subclusters. Predicted differences in CDKN2A RNA similarly validated the HPV status predictions, albeit the p-values were not as robust because of the smaller sample numbers in these cohorts.

## Assessing gene dropout

The phenomena whereby low sequencing depth randomly leads to genes not being represented is well described for single cell RNA-Seq (scRNA)experiments [53, 54]. We wondered whether gene dropout might also occur in FFPE samples for different biological or technical reasons, such as selective RNA instability. Put in mathematical terms, gene dropout means observing more samples with zero gene expression than expected by chance. We modeled the expected probability of observing samples with zero expression individually for every gene using the existing OPSCC TCGA cohort (N = 71) as a gold standard because the dataset was comparatively large and derived using frozen tissue. Because our cohort of samples were predicted to contain roughly 50% HPV-associated tumors, we used an average of the observed number of zero reads between HPV-associated and HPV-independent sample in the TCGA (**Supplementary Table 11** in S1 Table) to derive an estimate for each gene of the probability of getting zero reads. For every gene (excluding variable immunoglobulin or T cell receptor genes), the binomial probability distribution was applied to approximate the individual probability (p-value) and adjusted p-value (for multiple testing) of observing at least as many samples found in our cohorts with zero counts. The gene-wise adjusted p-values for all 3 of our RNA-seq cohorts are provided in the supplementary tables (**Supplementary Table 12**–14-**14** in S1 Table)**.** To compare gene dropout between sequencing platforms, we plotted the frequency distribution of genes against the log adjusted p-values (Fig 5A). Counting the number of genes with adjusted p-values < 0.01, dropout was highest in the TruSeq cohort (N = 3,100), lowest for the Stranded/Ribo-zero dataset (N = 583), and in between with the MACE-Seq samples (N = 2,181). A Venn diagram identifying overlap of gene dropout is shown in Fig 5B.

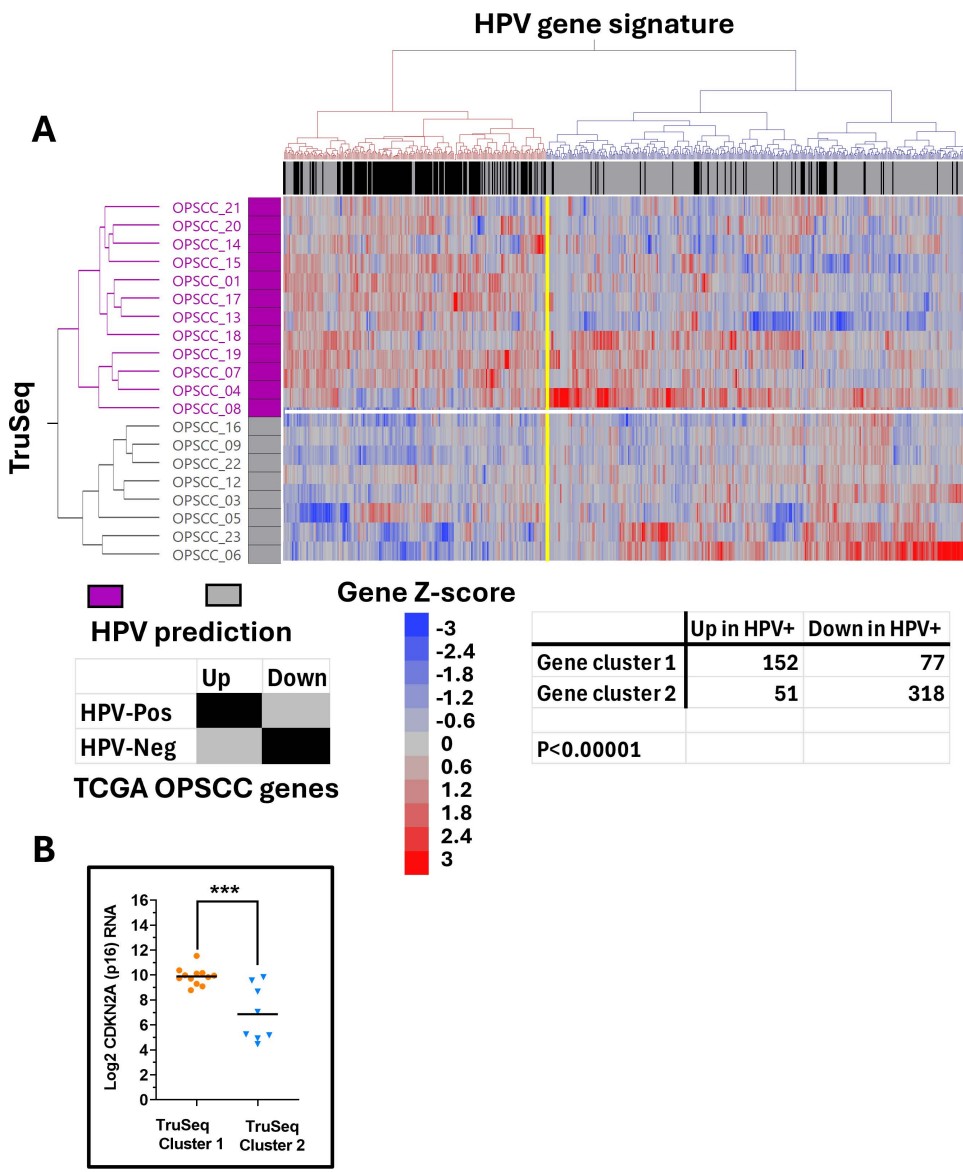

**Fig 3. Correlation of HPV associated genes in TruSeq cohort.** Expression of HPV-associated DEGs (identified from the TCGA OPSCC reference cohort) was used for unsupervised 2-way clustering of TruSeq samples (i.e., Ward's agglomerative hierarchical clustering) to predict HPV status (A). Samples predicted to be HPV-associated (i.e., HPV-Pos) are annotated with purple boxes. The regulation status of genes is annotated across the top of the heatmap with black boxes if they were also upregulated in HPV-associated TCGA samples or grey boxes if they were upregulated in HPV-independent (i.e., HPV-Neg) TCGA samples. Significant enrichment of genes upregulated in TCGA HPV-associated cancers was found in gene cluster 1 (red cluster) and enrichment of genes upregulated in HPV-independent TCGA samples is found in gene cluster 2 (blue cluster), demonstrating significant separation (e.g., P<0.00001 by Chi-square testing). Specimens from sample cluster 1 (purple) predicted to be HPV-associated based on their gene expression pattern had significantly higher CDKN2A (p16) expression (P) <0.0005 than specimens from sample cluster 2 (grey) predicted to be HPV-independent (B), validating predictions.

To better understand the study limitations or biological bias potentially introduced from gene dropout we performed a Gene Ontology Enrichment Analysis on the 283 common genes, and then for each cohort's dropout after removing these common genes. The 283 genes commonly missing from all cohorts were enriched for genes related to sexual reproduction

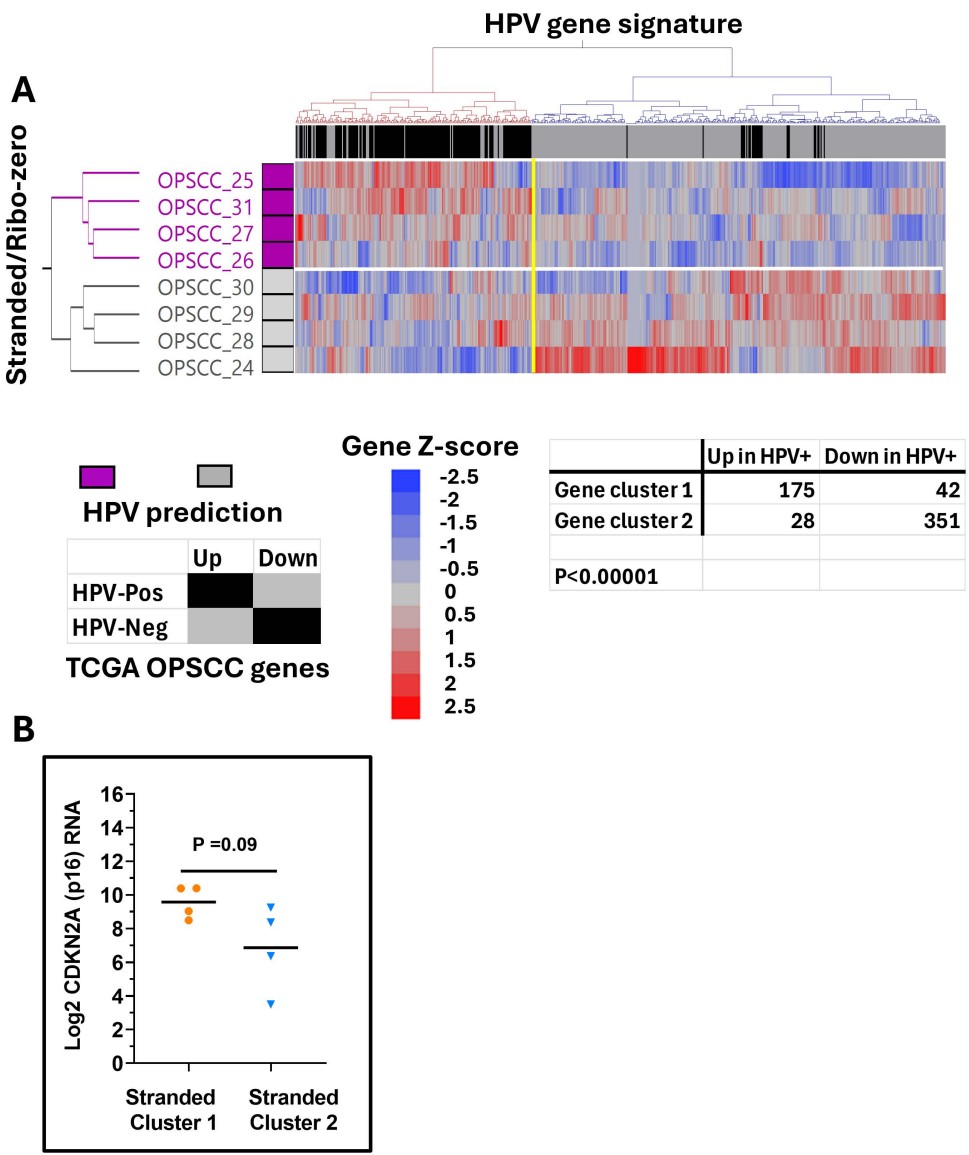

**Fig 4. Correlation of HPV associated genes in the Stranded Cohort.** Expression of HPV-associated DEGs (identified from the TCGA OPSCC reference cohort) was used for unsupervised 2-way clustering of Stranded cohort samples (i.e., Ward's agglomerative hierarchical clustering) to predict HPV status (A). Samples predicted to be HPV-associated (i.e., HPV-Pos) are annotated with purple boxes. The regulation status of genes is annotated across the top of the heatmap with black boxes if they were also upregulated in HPV-associated TCGA samples or grey boxes if they were upregulated in HPV-independent (i.e., HPV-Neg) TCGA samples. Significant enrichment of genes upregulated in TCGA HPV-associated cancers was found in gene cluster 1 (red cluster) and enrichment of genes upregulated in HPV-independent TCGA samples is found in gene cluster 2 (blue cluster), demonstrating significant separation (e.g., **P** < 0.00001 by Chi-square testing). Specimens from sample cluster 1 (purple) predicted to be HPV-associated based on their gene expression pattern had higher CDKN2A (p16) expression than specimens from sample cluster 2 (grey) predicted to be HPV-independent (B), which nearly reached significance.

and actually depleted for genes involved in DNA damage or metabolism (**Supplementary Table 15** in S1 Table). Implicit from the analysis of common gene dropout, genes associated with metabolism or nucleic acid regulation (e.g., synthesis, repair) were also statistically missing or not among the lists of gene dropout for all three sequencing platforms (**Supplementary**

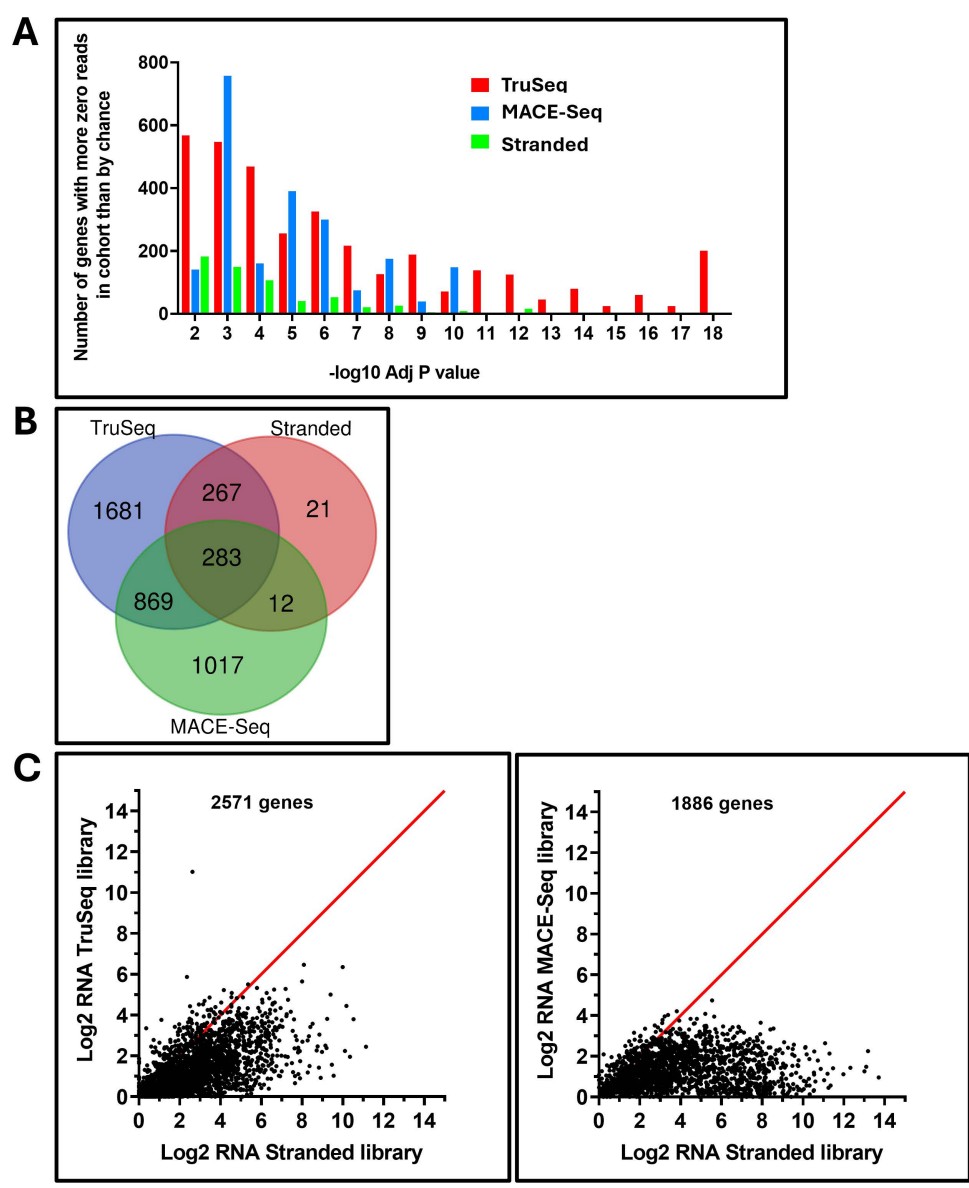

**Fig 5. Gene dropout.** Frequency distribution for number of genes verses the -log 10 Adj p-values derived from binomial probability calculations (A) demonstrated dropout was highest in the TruSeq cohort (N = 3,100), lowest for the Stranded/Ribo-zero dataset (N = 583), and in between with the MACE-Seq samples (N = 2,181); overlapping dropout genes (B). C) Cohort medians for genes with dropout in the TruSeq or MACE-seq dataset plotted against their medians in the Stranded/Ribo-zero dataset demonstrate a bias towards dropout for low expression genes.

Table 16–18-18 in S1 Table). Genes with higher-than-expected loss from both the TruSeq and MACE-Seq cohorts were enriched for neuro-regulation pathways (**Supplementary Tables 16-17** in S1 Table), which might be attributed to a higher inclusion of adjacent nerve cells in the TCGA cohort—possibly due to larger average size of specimens in the latter. Although far fewer in number, genes with dropout from the Stranded/Ribo-zero dataset were enriched for keratinization, which could be attributed to differences in the cohort makeup given the small sample size and high variability of differentiation markers naturally present among OPSCC tumors. To provide insight into the differences in gene dropout between samples prepared

using the Stranded/Riob-zero kit, we plotted the cohort medians for genes with dropout in the TruSeq or MACE-seq dataset against their medians in the Stranded/Ribo-zero dataset (Fig 5C). Genes that dropped out had a bias towards lower expression in the Stranded/Ribo-zero dataset and this was most prominent when examining gene dropout in the TruSeq cohort.

Unlike scRNA, where dropout events are randomly distributed among samples, missing genes in the FFPE analysis were consistently absent across all samples within a given platform—though variability was observed between platforms. To better understand the source(s) of these differences, we compared the expected expression of genes well represented in the three platforms to the expected expression of genes with significant dropout using their average expression values derived from the TCGA OPSCC cohort, computed after equally weighting HPV-independent and HPV-associated samples (S10 Fig)- Compared to the expected distribution of all genes (S10 Fig A), the 283 common genes to dropout from all platforms and those specifically missing from the TruSeq or Stranded libraries were biased towards low expression (S10 FigB–D), whereas the 1017 genes specifically absent from the MACE-Seq library preparation were more evenly distributed across all levels of expression (S10 FigE). Underrepresentation of GC rich regions have been reported when RNA sequencing is performed from fixed specimens [55] possibly because GC rich regions can form more stable secondary structures, increasing the frequency of formalin cross linking. Indeed, genes with dropout were found to have on average significantly higher GC content when compared to genes that were well represented (S10 Fig F and G) among all three library platforms, but the differences were greatest among the common dropout genes (51% GC) and those missing from Stranded library preparation (51%) compared to well represented genes which ranged from 46% to 47% GC. However, genes missing from the MACE-Seq library preparation were less impacted by GC content. Collectively, the analysis suggests a bias in dropout towards genes with low expression and higher GC content when RNA is prepared using the TrueSeq or Stranded libraries. However, MACE-Seq captures only the 3' end of transcripts and relies more heavily on the presence of poly(A) tails, which are susceptible to degradation during fixation [56]. This degradation may explain the distinct pattern of gene dropout observed.

### GC-bias among different library platforms

Because GC content was one of the components influencing gene dropout, we examined how it impacted expression more broadly. For each library preparation, we plotted the average log2 gene expression values against the % GC content of the canonical gene transcripts and compared the smoothed curves to average values derived from the TCGA OPSCC cohort as a gold standard. Comparisons were made after stratifying samples according to their known (TCGA) or predicted HPV status. For TCGA OPSCC samples, regardless of HPV status, there was a slight GC-bias with peak gene expression occurring at about 50% GC content, with average expression gradually declining as the % GC content increased further (S11 Fig A and B). Compared to the TCGA, the GC-bias (i.e., decline in expression with increasing % GC) was most apparent when samples were prepared using the TruSeq library compared to samples analyzed from the Stranded or MACE-Seq libraries (S11 Fig A–D), with no observed differences based on predicted HPV status. Multiple bioinformatic approaches exist to correct for GC-bias in RNAseq data. Therefore, we developed an analytical tool called "GCplotQC" to facilitate identification and quantification of GC-bias among specific samples to guide decisions pertaining to whether these GC correction techniques would be appropriate for experimental data. The program utilizes expression data from each sample along with known GC content for canonical transcripts to overlay plots of expression against % GC for the sample and a reference (S12 Fig), which could either be derived from the average of samples or a comparator like the

TCGA. The program then calculates the area of the sample curve beneath the comparator reference curve (i.e., deviation), which is displayed on the plot and conveniently aggregated in a master table. Although not specifically driven by GC-bias, technical outliers identified earlier by our pipeline were easily distinguished from other samples by their much larger deviations (S12 Fig B, D, and F). When examined individually, the average deviations for samples prepared with the TruSeq library were significantly higher then those derived from samples prepared using the Stranded or MACE-Seq library approach (S12 Fig G).

**Co-extraction of DNA and SNP profiling patient ancestry**

We analyzed DNA pellets left over during RNA extraction from a subset of samples using a modified COBAS protocol for ancestry using the Illumina Global Diversity SNP array. Our average DNA yield was 245 ng or 24 ng/cm$^2$ of tissue (**Supplementary Table 19** in S1 Table), which was sufficient for the SNP platform where 200 ng is the recommended amount. For comparison we isolated DNA from additional specimens that were not processed for RNA and ran them on the same SNP array. We found that the leftover DNA extracted during the RNA isolations performed satisfactorily but had slightly diminished SNP call rates which averaged 83.5% compared to 92.0% for samples where only DNA was isolated (P = 0.05, Fig 6A). In all cases, we were able to use the SNP data generated to predict ancestry (**Supplementary Table 20** in S1 Table) using software freely available. Using the K18M model of SNPs, we identified a variety of ethnic groups at various percentages, but 9 out of 14 (e.g., 64%) were of Atlantic/Northern European descent, which was consistent with patient demographics for OPSCC. Two patients were found to be African American, and another was of Western Mediterranean ancestry. In samples from patients #1 (OPSCC_03/OPSCC_26) and patient # 8 (e.g., OPSCC_24), we isolated DNA directly or after RNA purification and compared their ancestry prediction. For both patients, (Fig 6B, 6C), the predicted ancestries matched accordingly, with one showing strong descent from Equatorial Africa and the other a mixed ancestry predominately of Atlantic/Northern European origin.

## Conclusions

In the present work we demonstrate the feasibility of utilizing archival FFPE specimens of varying age for whole transcriptome sequencing (i.e., RNA-Seq) to obtain biologically meaningful data. Further, we provide technical details for: 1) isolating both RNA and DNA, 2) insight into the most meaningful quality control metrics, 3) methods for standardizing data analysis, and 4) suggestions for data quality control increasing confidence in the data. Lastly, we show that residual DNA recovered from specimens during RNA isolation is of sufficient quality for SNP array platforms, which can be used to confirm ancestry or racial identity of study participants. We recommend the AllPrep DNA/RNA FFPE kit to isolate RNA because in our hands it performed robustly with respect to yield for nearly 200 specimens and led to high quality downstream RNA-seq reads using three different library prep methods. On average, a total of 18 cm$^2$ of tissue (e.g., less than 10 slides) cut at standard thickness (6 μM) generated more than sufficient RNA for the different sequencing options available. If RNA yields are less than 246 ng/cm$^2$ (**Supplementary Table 2** in S1 Table, lower 90% confidence interval of mean) based on A260 measurements, then appropriate trouble shooting steps are warranted. FFPE RNA is known to be fragmented, so typical metrics like RIN are not appropriate. DV200 measurements should be interpreted cautiously, because it is still possible to generate reliable RNA-Seq data from samples with a DV200 < 20%. Rather the shape of the TapeStation curve closer to 100 bp may be a better indicator.

After filtering out non-protein coding genes (e.g., **Supplementary Table 4** in S1 Table) our quality control process consists of five main steps easily applied, followed by some type of analysis for biological fidelity that by necessity will be context specific. We have incorporated

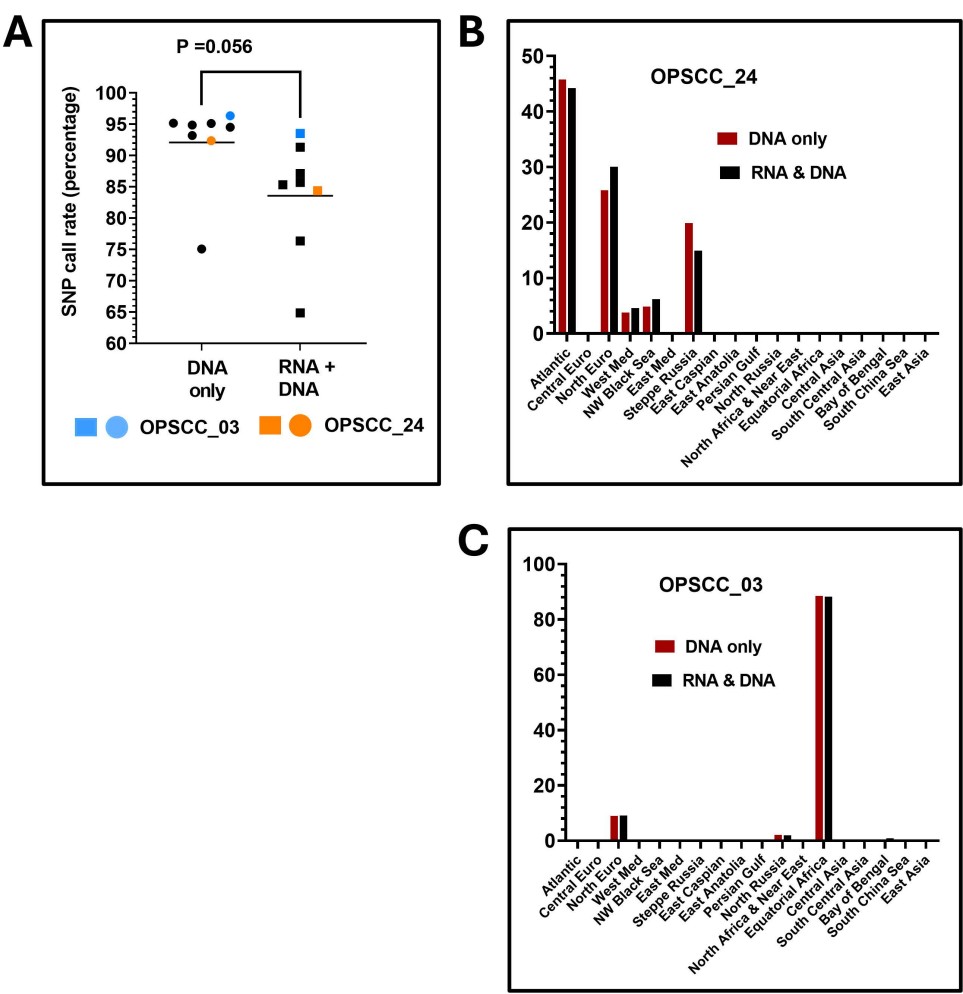

**Fig 6. Ancestry measurements.** A) SNP call rate was calculated for each DNA extraction approach for each individual sample. B, C) Comparison of ancestry results from two unique samples where DNA was extracted twice using either the COBAS method alone (red) or from residual pellets following RNA isolation in a combined procedure (black).

an UQ normalization into our ready to use Python code (**Supplementary Methods**), but other methods could work as well. The percentage of mapped reads belonging to protein coding genes should be examined for each sample with the caveat that stranded library approaches may lower the values because of the additional requirement that captured fragmented DNA also be from a specific strand. Our method of detecting technical outliers based on MAD residuals works differently than principal component analysis (PCA) and should be applied upstream of the latter [57]. Unlike PCA, this method makes use of information from every gene rather than a few thousand most variable genes. Therefore, the MAD method may have better sensitivity for detecting just a single outlier in larger cohorts where the most variable genes selected for PCA may be driven largely by biological variability. In fact, it is probably more consistent than PCA when it comes to separating technical rather than biologic variation. However, the method is not a substitute for PCA that could be applied afterwards to resolve differences based on biologic variation that may reveal further heterogeneity to be considered when comparing gene expression.

The number of protein coding genes with log2 RNA expression ≥ 2 should be calculated only after re-adjusting the global median to equal 7. This re-centering allows the same standard threshold for low gene expression to be more consistently applied, regardless of sequencing depth, batch affects, or platforms used. By way of comparison, the OPSCC TCGA cohort samples average 14,565 genes (Range:13,383–15,389) with log2 expression ≥ 2. Typically, samples with too few protein coding genes above the low expression threshold are easily identified as technical outliers. Using the MAD residuals method avoids having to set a precise cut-off value *for number of genes* above the threshold to consider acceptable, provided most samples in the cohort express reasonable numbers of genes. For our datasets, excluding outliers, the average number of genes above the threshold of 2 was ~14,000. It is useful to compare the distribution (i.e., histogram) of normalized log2 RNA expression values to a standard like the TCGA data, which follows a quasi-log normal distribution. In our experience, the re-centered RNA-seq data from the TCGA has a similar appearance regardless of the cancer chosen.

In one of the last simple QC steps, we recommend identifying genes with an abnormal degree of sample dropout, if the expected frequency of samples with zero expression can be appropriately modeled with an orthogonal data set. We observed the least amount of gene dropout in the cohort prepared using the Stranded library kit, however, it is likely that the average quality of RNA was better for these specimens judging by the TapeStation curves. Regardless, for the TruSeq and Stranded datasets the genes lost seemed unrelated to tumor biology and likely represented low expression genes However, gene dropout out in the MACE-Seq cohort was independent of expression level, and likely reflects higher instability of RNA at the poly A tail. Nevertheless, genes related to metabolism or nucleic acid processing were depleted from the lists of dropout in all cohorts and therefore were reasonably represented.

Although our cohort sizes were limited, we detected substantially more GC-bias among samples prepared with the TruSeq library. The GCplotQC tool we developed could be useful to detect samples with unusual GC-bias that may stem from over fixation. Although our RNA isolation procedure includes an incubation step at elevated temperatures to reverse formalin cross-links, samples that are over fixed would be predicted to preferentially lose expression of RNA transcripts with higher GC content. Higher GC content is known to stabilize RNA secondary structure thereby increasing the odds of formalin cross-linking. Compared to the TCGA cohort, which utilized frozen specimens, our FFPE samples did exhibit decreased RNA expression with increasing % GC content. However, the deviations we observed were more uniform across a range of GC content and samples, suggesting a more generalized effect inherent from the formalin fixation that was minimally present with samples from the MACE-Seq and Stranded cohorts. The GC-bias present with TruSeq samples was significantly more pronounced. A variety of bioinformatic approaches to correct for GC-bias in RNAseq data exist [58,59], including methods that perform adjustment during alignment (e.g., BBMAp from BBTools), before quantitation (e.g., CorrectGCBias from RseQC), or after quantitation (e.g., RUVSeq). We recommend using the GCplotQC tool to gauge the extent of GC-bias among the entire cohort and individual samples, preferably using TCGA data as a reference. Individual samples that suffer from over fixation may stand out as technical outliers using our pipeline or possibly have significantly fewer genes expressed above our recommended log2 threshold of 2.0. Alternatively, cases where the impact of over fixation is more subtle would likely be apparent as a sharp decline in gene expression at some %GC threshold, which could be visualized through our GCplotQC tool.

If possible, the biological fidelity of new data should be evaluated independent of the scientific questions being explored. For our data set, we leveraged the fact that a substantial subset of OPSCCs are linked to high risk HPV, which leads to programs of genes that are

both up and downregulated. We then used hierarchical clustering to examine if these genes defined by an independent cohort (i.e., TCGA data) behaved similarly in our datasets. This same approach could be applied using just about any set of genes from pathways known to be co-regulated in the tissue under study. Co-regulation or anti-correlations among gene clusters can be examined statistically using simple Chi-square tests. Lastly, we showed that SNP data from residual DNA, co-isolated during RNA purification, is of sufficient quality for SNP arrays and can be expected to have call rates that average above 80%, which may be sufficient for many applications and gives comparable results to DNA directly isolated from FFPE. Our data, methods, and recommended analytical approaches should provide investigators with helpful guidelines and tools to better leverage archival specimens to answer important biological questions.

## Supporting information

**S1 Fig. RNA quantitation.** RNA concentrations measured via fluorescence compared to internal laboratory measures using optical density (OD).
(TIF)

**S2 Fig. Analytical pipeline.** Pipeline for analysis of RNA-seq data inclusive of normalization.
(TIF)

**S3 Fig. Coding vs non-coding reads.** Quantification of coding vs non-coding reads mapped using the True-Seq and Stranded RNA with Ribo-zero platforms (A) and the MACE RNAseq platform (B). Head to head comparison across platforms (C). Sample OPSCC_33, which had the lowest % of protein coding reads (red circle) was also a technical outlier (Supplementary Table 5 in S1 Table).
(TIF)

**S4 Fig. Distribution of RNA median gene expression.** The distributions of gene expression (i.e., median UQ normalized log 2 values) for each platform/cohort, along with the TCGA OPSCC RNA-Seq dataset similarly normalized (A,C,E,G) before global rescaling. Histograms for the rescaled cohorts were also generated (B,D,F,H).
(TIF)

**S5 Fig. Average Median Absolute Deviation (MAD).** MAD values were averaged across all genes to calculate a sample specific average MAD value. Samples OPSCC_32 and 33 (Stranded RNA/Ribo-zero), OPSCC_34 and 35 (MACE-Seq) were identified as outliers with significantly different MAD values (Supplementary Table 5 in S1 Table).
(TIF)

**S6 Fig. Relationship between DV200 and biological richness of gene expression.** DV200 values and RNA fragment size are shown for representative samples along with the relative number of usable genes and associated p-values from outlier analysis (Supplementary Table 5 in S1 Table).
(TIF)

**S7 Fig. Correlation of HPV-associated genes and CDKN2A expression in TCGA samples.** A) Genes that showed ≥3-fold significant (FDR < 0.1) difference up or downregulation based on HPV status in the OPSCC TCGA cohort were identified. B) Confirmation that CDKN2A expression is highly upregulated in HPV-associated (HPV-pos) TCGA samples. ****P < 0.0001.
(TIF)

**S8 Fig. Cross-correlation of HPV associated genes.** Cross-correlation coefficients of gene expression values within the TruSeq cohort, using the list of 855 DEGs previously associated with HPV status in the TCGA OPSCC samples, were used for unsupervised clustering to identify modules of genes (black boxes) that behaved similarly. Gene clusters 1 and 3 behaved robustly. Genes are annotated vertically and horizontally by whether they were upregulated (red boxes) or downregulated (grey boxes) in the original TCGA cohort according to HPV status. (TIF)

**S9 Fig. Correlation of HPV associated genes in the MACE-Seq cohort.** Expression of HPV-associated DEGs (identified from the TCGA OPSCC reference cohort) was used for unsupervised 2-way clustering of MACE-Seq cohort samples (i.e., Ward's agglomerative hierarchical clustering) to predict HPV status (A). Samples predicted to be HPV-associated (i.e., HPV-Pos) are annotated with purple boxes. The regulation status of genes is annotated across the top of the heatmap with black boxes if they were also upregulated in HPV-associated TCGA samples or grey boxes if they were upregulated in HPV-independent (i.e., HPV-Neg) TCGA samples. Significant enrichment of genes upregulated in TCGA HPV-associated cancers was found in gene cluster 1 (red cluster) and enrichment of genes upregulated in HPV-independent TCGA samples is found in gene cluster 3 (dark blue cluster), demonstrating significant separation (e.g., $P < 0.00001$ by Chi-square testing). Specimens from sample cluster 1 (purple) predicted to be HPV-associated based on their gene expression pattern had higher CDKN2A (p16) expression than specimens from sample cluster 2 (grey) predicted to be HPV-independent (B). * $P < 0.05$. (TIF)

**S10 Fig. Sources of gene dropout.** A) Distribution of HPV-associated and HPV-independent weighted averages of RNA expression for all genes using the TCGA OPSCC cohort. B) Distribution of average expression from the TCGA OPSCC cohort for the 283 common dropout genes across all FFPE library platforms. C) Distribution of average expression from the TCGA OPSCC cohort for the 1681 genes that uniquely dropped out from the TruSeq platform. D) Distribution of average expression from the TCGA OPSCC cohort for the 1017 genes that uniquely dropped out from the MACE-Seq platform. E) Distribution of average expression from the TCGA OPSCC cohort for the 248 genes (21 + 267) that uniquely dropped out from the Stranded and/or TruSeq platforms. F) Distribution of GC content among genes that were covered well or dropped out for each platform. G) Statistical comparisons between genes that dropped out and those well covered for each of the platforms or the common 283 dropout genes. The P-values (**** $P < 0.00001$, *** $P < 0.0005$) were derived from individual comparisons of averages connected by solid lines after a Tukey's multiple comparison test. (TIF)

**S11 Fig. GC-bias in RNA-Seq data from FFPE samples.** A) Smoothed spline plots comparing average gene expression verses % GC content of canonical transcripts for HPV-independent FFPE samples prepared using the TruSeq (dark blue line) or Stranded (light blue line) libraries compared to HPV-independent TCGA OPSCC samples (black line). B) Smoothed spline plots comparing average gene expression verses % GC content of canonical transcripts for FFPE samples predicted to be HPV-associated and prepared using the TruSeq (dark blue line) or Stranded (light blue line) libraries compared to HPV-associated TCGA OPSCC samples (black line). C) Smoothed spline plots comparing average gene expression verses % GC content of canonical transcripts for FFPE samples predicted to be HPV-independent and prepared using the MACE-Seq approach (red line) compared to HPV-independent TCGA OPSCC samples (black line). D) Smoothed spline plots comparing

average gene expression verses % GC content of canonical transcripts for FFPE samples predicted to be HPV-associated and prepared using the MACE-Seq approach (red line) compared to HPV-associated TCGA OPSCC samples (black line).
(TIF)

**S12 Fig. Qualitative and quantitative assessment of GC bias using the GCplotQC tool.** Smoothed spline curves of gene expression verses % GC content of canonical transcripts for individual samples processed using the TruSeq library (A,B), the Stranded library (C,D), or the MACE-Seq library (E,F) compared to a weighted average of HPV-associated and HPV-independent OPSCC TCGA samples for reference. The program computes the area between the reference (OPSCC TCGA samples) and individual sample curves, wherever the sample curve has lower gene expression and displays the value as delta area. Samples identified as technical outliers (B,D, and F) in other steps of the pipeline also show dramatic increases in delta area. G) Scatter plot of average sample delta area values shows significantly more deviation in samples prepared with the TruSeq library compared to those prepared with the other two libraries. **** $P < 0.00001$.
(TIF)

**S1 Table. Supplemental Tables.**
(XLSX)

## Acknowledgements

Not applicable.

## Author contributions

**Conceptualization:** Vlad Sandulache, Patricia Castro, Thomas J. Ow, Heath D. Skinner.

**Data curation:** Mitchell J. Frederick, Dannelys Perez-Bello, Alan Frederick, Andrew Frederick, Rashid A. Osman, Fonma Essien, Imelda Yebra, Ashley Hamlin.

**Formal analysis:** Mitchell J. Frederick, Dannelys Perez-Bello, Pedram Yadollahi, Patricia Castro, Alan Frederick, Andrew Frederick.

**Funding acquisition:** Vlad Sandulache, Heath D. Skinner.

**Investigation:** Vlad Sandulache, Mitchell J. Frederick, Pedram Yadollahi, Andrew Frederick, Rashid A. Osman, Fonma Essien, Imelda Yebra, Ashley Hamlin.

**Methodology:** Vlad Sandulache, Mitchell J. Frederick, Dannelys Perez-Bello, Pedram Yadollahi, Patricia Castro, Rashid A. Osman, Fonma Essien, Imelda Yebra, Ashley Hamlin, Thomas J. Ow, Heath D. Skinner.

**Project administration:** Vlad Sandulache, Mitchell J. Frederick, Dannelys Perez-Bello, Pedram Yadollahi.

**Resources:** Vlad Sandulache, Patricia Castro.

**Software:** Mitchell J. Frederick, Alan Frederick, Andrew Frederick, Rashid A. Osman.

**Supervision:** Vlad Sandulache, Mitchell J. Frederick, Dannelys Perez-Bello.

**Validation:** Mitchell J. Frederick, Pedram Yadollahi, Alan Frederick, Thomas J. Ow.

**Writing – original draft:** Vlad Sandulache, Mitchell J. Frederick, Dannelys Perez-Bello, Thomas J. Ow, Heath D. Skinner.

**Writing – review & editing:** Vlad Sandulache, Mitchell J. Frederick, Dannelys Perez-Bello, Pedram Yadollahi, Patricia Castro, Alan Frederick, Andrew Frederick, Rashid A. Osman, Fonma Essien, Imelda Yebra, Ashley Hamlin, Thomas J. Ow, Heath D. Skinner.

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
