## [Decision Letter · Decision Letter 0]

13 Jan 2025

PONE-D-24-53608Reliable RNA-seq analysis from FFPE specimens as a means to accelerate cancer-related health disparities researchPLOS ONE

Dear Dr. Sandulache,

Thank you for submitting your manuscript to PLOS ONE. After careful consideration, we feel that it has merit but does not fully meet PLOS ONE’s publication criteria as it currently stands. Therefore, we invite you to submit a revised version of the manuscript that addresses the points raised during the review process.

We look forward to receiving your revised manuscript.

Kind regards,

Kazunori Nagasaka

Academic Editor

PLOS ONE

Journal Requirements:

 “VCS- R21DE032344

HDS- R01DE0323337; R01 DE028061”

6. PLOS requires an ORCID iD for the corresponding author in Editorial Manager on papers submitted after December 6th, 2016. Please ensure that you have an ORCID iD and that it is validated in Editorial Manager. To do this, go to ‘Update my Information’ (in the upper left-hand corner of the main menu), and click on the Fetch/Validate link next to the ORCID field. This will take you to the ORCID site and allow you to create a new iD or authenticate a pre-existing iD in Editorial Manager.

7. We note that you have included the phrase “data not shown” in your manuscript. Unfortunately, this does not meet our data sharing requirements. PLOS does not permit references to inaccessible data. We require that authors provide all relevant data within the paper, Supporting Information files, or in an acceptable, public repository. Please add a citation to support this phrase or upload the data that corresponds with these findings to a stable repository (such as Figshare or Dryad) and provide and URLs, DOIs, or accession numbers that may be used to access these data. Or, if the data are not a core part of the research being presented in your study, we ask that you remove the phrase that refers to these data.

8. We note you have not yet provided a protocols.io PDF version of your protocol and/or a protocols.io DOI. When you submit your revision, please provide a PDF version of your protocol as generated by protocols.io (the file will have the protocols.io logo in the upper right corner of the first page) as a Supporting Information file. The filename should be S1_file.pdf, and you should enter “S1 File” into the Description field. Any additional protocols should be numbered S2, S3, and so on. Please also follow the instructions for Supporting Information captions [https://journals.plos.org/plosone/s/supporting-information#loc-captions]. The title in the caption should read: “Step-by-step protocol, also available on protocols.io.”

Please assign your protocol a protocols.io DOI, if you have not already done so, and include the following line in the Materials and Methods section of your manuscript: “The protocol described in this peer-reviewed article is published on protocols.io (https://dx.doi.org/10.17504/protocols.io.[...]) and is included for printing purposes as S1 File.” You should also supply the DOI in the Protocols.io DOI field of the submission form when you submit your revision.

If you have not yet uploaded your protocol to protocols.io, you are invited to use the platform’s protocol entry service [https://www.protocols.io/we-enter-protocols] for doing so, at no charge. Through this service, the team at protocols.io will enter your protocol for you and format it in a way that takes advantage of the platform’s features. When submitting your protocol to the protocol entry service please include the customer code PLOS2022 in the Note field and indicate that your protocol is associated with a PLOS ONE Lab Protocol Submission. You should also include the title and manuscript number of your PLOS ONE submission.

Additional Editor Comments:

Dear Authors,

Thank you for submitting your manuscript to PLOS One.

Based on the comments from our reviewers, our decision is "Minor Revision." Please revise the manuscript accordingly, and we look forward to receiving your revised submission.

Sincerely,

Kazunori Nagasaka

Reviewers' comments:

Reviewer's Responses to Questions

**Comments to the Author**

1. Does the manuscript report a protocol which is of utility to the research community and adds value to the published literature?

Reviewer #1: Yes

Reviewer #2: Yes

2. Has the protocol been described in sufficient detail?

To answer this question, please click the link to protocols.io in the Materials and Methods section of the manuscript (if a link has been provided) or consult the step-by-step protocol in the Supporting Information files.

The step-by-step protocol should contain sufficient detail for another researcher to be able to reproduce all experiments and analyses.

Reviewer #1: Yes

Reviewer #2: Yes

3. Does the protocol describe a validated method?

Reviewer #1: Yes

Reviewer #2: No

4. If the manuscript contains new data, have the authors made this data fully available?

Reviewer #1: Yes

Reviewer #2: Yes

**5. Is the article presented in an intelligible fashion and written in standard English?**

Reviewer #1: Yes

Reviewer #2: Yes

6. Review Comments to the Author

Reviewer #1: In this submitted protocol authors optimized guidelines for DNA and RNA purification from FFPE oropharynx carcinoma tissue, followed by RNA-seq and DNA SNP analysis.

Introduction

The rational background of the protocol establishment, the given requirements are properly described in the Introduction.

Results

The provided protocol with the details included in supplementary documents is a gap-filling requirement. The presented results are well-summarized and logically described. A correct and critical comparison to TCGA data is provided, discussed, and reasons for cross-dataset correlations failures were explained.

Discussion

The Discussion highlights the main points in this lab method process and provides suggestions.

This submitted material is a great contribution to the research in the corresponding field, but also valuable for interdisciplinary use.

Reviewer #2: Large collection of FFPE archival biopsies from hospitals and research institutes are valuable samples for generating transcriptome and exome data. The authors have provided a detailed protocol to isolate RNA and DNA from FFPE samples, generate RNA-seq and SNP array data, and perform bioinformatics to remove outlier samples and normalize expression levels between samples. Although they do not develop any new methods, the authors provide an end-to-end protocol from RNA/DNA isolation to bioinformatics analysis by combining existing methods with some modifications. The protocol may be useful to researchers without prior experience working with FFPE samples.

Major deficiency of the described bioinformatics method is that it does not take into consideration the impact of the chemistry of formalin fixation on the sequences. It is well known that formalin fixation primarily causes modification of amino groups on the four bases (A, C, G, U) with differing probabilities and can impact the GC content of the sequenced reads. Also, the decreased gene coverage in RNA-seq (described as dropouts by the authors) is not completely random and is very different from the dropouts observed in single-cell RNA-seq. Thus, it is essential that QC checks specific to FFPE, especially formalin overfixation, is necessary. The method described by the authors may work when majority of the samples produce good quality data and a small proportion of outlier samples need to be identified, but not when a large proportion of samples have QC issues due to fixation protocol or FFPE storage issues. Please include additional QC steps that are specific to FFPE samples and steps to identify formalin overfixation.

7. PLOS authors have the option to publish the peer review history of their article (what does this mean? ). If published, this will include your full peer review and any attached files.

**Do you want your identity to be public for this peer review?** For information about this choice, including consent withdrawal, please see our Privacy Policy .

Reviewer #1: **Yes: ** Jozsef Dudas

Reviewer #2: No

---

## [Author Response · Author response to Decision Letter 1]

17 Feb 2025

Response to reviewer comments:

Reviewer #1: This submitted material is a great contribution to the research in the corresponding field, but also valuable for interdisciplinary use.

Response: Thank you.

Reviewer #2: Major deficiency of the described bioinformatics method is that it does not take into consideration the impact of the chemistry of formalin fixation on the sequences. It is well known that formalin fixation primarily causes modification of amino groups on the four bases (A, C, G, U) with differing probabilities and can impact the GC content of the sequenced reads. Also, the decreased gene coverage in RNA-seq (described as dropouts by the authors) is not completely random and is very different from the dropouts observed in single-cell RNA-seq. Thus, it is essential that QC checks specific to FFPE, especially formalin overfixation, is necessary. The method described by the authors may work when majority of the samples produce good quality data and a small proportion of outlier samples need to be identified, but not when a large proportion of samples have QC issues due to fixation protocol or FFPE storage issues. Please include additional QC steps that are specific to FFPE samples and steps to identify formalin overfixation.

Response: We appreciate the astute observation and suggestions and agree completely. We now include a newly created tool called “GCplotQC” developed to analyze GC-bias in individual samples and quantitatively compare deviation from a reference standard like TCGA data or the cohort average. The tool graphically generates smoothed curves from plots of %GC content verses gene expression and calculates the area of deviation compared to a designated standard reference. The area of deviation for all samples is then outputted to a table for easy reference, while the curves for individual samples are stored as png images and aggregated as a PDF file. This can be used to gauge overall GC bias among the entire cohort or individually identify samples that have steeper GC bias presumably stemming from over fixation. We have now added two paragraphs to the Results section that describe the impact of GC-bias in our study, how that relates to gene dropout, and the differences we observed between the three library preparations used. These results are illustrated in three brand new supplementary figures now added (Supplementary Figures 10, 11, and 12). We also added an additional paragraph to the Discussion section that outlines the utility of the QC tool, and references bioinformatic techniques developed by others that could be applied to correct for GC-bias at various steps in the analysis.

To more specifically answer the reviewer’s questions, we agree that the zero dropout we observed was not through the same mechanisms found for single cell RNAseq because the latter impacts samples randomly and our zero dropouts occurred uniformly across samples. This is now explained better in our modified Results section. We find that zero dropout in our cohorts was driven by both low expected gene expression and to some degree GC content. However, in our case the GC bias was more likely systemic and uniform across samples within a cohort that likely reflects a general limitation of formalin fixation. Indeed, this is why our sample processing protocol includes an elevated temperature incubation to reverse and mitigate the effects of fixation. We report that the GC-bias was much more pronounced for the TruSeq samples than for those prepared with the Stranded or MACE-Seq libraries. As for how much GC-bias is too much before an analytical correction should be applied, we leave that to the investigator to determine for themselves. Our tool will allow quantitation and visualization of the affect (e.g. QC), which can be used in decisions governing whether to utilize one of the many GC correction pipelines already available. The use of a reference standard like TCGA data should help detect whether some samples are over fixed, or the entire cohort is bad, and our methods employ a quantitative metric derived from the area between a sample’s curve and the curve generated from a chosen standard. We are very grateful for the reviewer’s challenging question and believe addressing the concern has significantly improved our manuscript and the utility of our published pipeline.

---

## [Decision Letter · Decision Letter 1]

10 Mar 2025

Reliable RNA-seq analysis from FFPE specimens as a means to accelerate cancer-related health disparities research

PONE-D-24-53608R1

Dear Dr. Sandulache,

We’re pleased to inform you that your manuscript has been judged scientifically suitable for publication and will be formally accepted for publication once it meets all outstanding technical requirements.

Kind regards,

Kazunori Nagasaka

Academic Editor

PLOS ONE

Additional Editor Comments (optional):

Dear Authors,

Thank you very much for submitting your revised manuscript. I am pleased to inform you that, after careful consideration, our expert reviewers have recommended your paper for acceptance in PLOS ONE.

We believe your research makes a clear and valuable contribution to the clinical field, and we anticipate that your findings will be widely recognized and cited in future studies.

Congratulations once again!

We look forward to seeing your published work and hope you continue your excellent research.

Sincerely,

Kazunori Nagasaka

Reviewers' comments:

Reviewer's Responses to Questions

**Comments to the Author**

1. Does the manuscript report a protocol which is of utility to the research community and adds value to the published literature?

Reviewer #2: Yes

2. Has the protocol been described in sufficient detail?

To answer this question, please click the link to protocols.io in the Materials and Methods section of the manuscript (if a link has been provided) or consult the step-by-step protocol in the Supporting Information files.

The step-by-step protocol should contain sufficient detail for another researcher to be able to reproduce all experiments and analyses.

Reviewer #2: Yes

3. Does the protocol describe a validated method?

Reviewer #2: Yes

4. If the manuscript contains new data, have the authors made this data fully available?

Reviewer #2: Yes

**5. Is the article presented in an intelligible fashion and written in standard English?**

Reviewer #2: Yes

6. Review Comments to the Author

Reviewer #2: Authors have updated the manuscript to address the GC bias caused by formalin fixation. Although the reason for the GC bias is not adequately or accurately described, the authors describe it more as observations and should be adequate for the protocol reporting. Authors have provided code and ways to potentially detect or minimize GC bias.

7. PLOS authors have the option to publish the peer review history of their article (what does this mean? ). If published, this will include your full peer review and any attached files.

**Do you want your identity to be public for this peer review?** For information about this choice, including consent withdrawal, please see our Privacy Policy .

Reviewer #2: No

---

## [Editor Report · Acceptance letter]

PONE-D-24-53608R1

PLOS ONE

Dear Dr. Sandulache,

I'm pleased to inform you that your manuscript has been deemed suitable for publication in PLOS ONE. Congratulations! Your manuscript is now being handed over to our production team.

Kind regards,

on behalf of

Professor Kazunori Nagasaka

Academic Editor

PLOS ONE